# Incremental diagnostic yield of consecutive sputum nucleic acid amplification tests for pulmonary tuberculosis

Takumi Kanokogi,[1,2] Naohisa Urabe,[1,2] Nozomi Tokita,[1] Hinako Murakami,[3] Masakazu Sasaki,[3] Susumu Sakamoto,[1,2] Kazuma Kishi[1,2]

**ABSTRACT** The incremental diagnostic yield of consecutive sputum nucleic acid amplification tests (NAATs) for pulmonary tuberculosis (PTB) is not well established. We aimed to assess the diagnostic performance of one to three consecutive sputum NAATs in a non-high-tuberculosis-burden setting. We retrospectively analyzed data from 4,051 patients with suspected PTB in Japan between 2010 and 2023. Patients with a positive culture or a positive NAAT consistent with clinical findings were classified into the PTB group ($n = 290$); all others were classified into the non-PTB group ($n = 3,761$). We evaluated the cumulative sensitivity and specificity of smear microscopy, NAAT, and culture for one, two, and three sputum specimens and compared the time to microbiological diagnosis for culture alone versus culture combined with one, two, or three sputum NAATs. In the PTB group, a second NAAT substantially increased cumulative sensitivity from 53.1% to 63.1% (+10.0%) and shortened the mean time to diagnosis from 10.9 to 8.3 days (−2.6 days). A third NAAT provided minimal additional benefit, increasing sensitivity to 67.2% (+4.1%) and reducing the time to diagnosis to 8.1 days (−0.2 days). In smear-positive PTB, the NAAT sensitivity increased from an already high 84.9% with a single test to 97.8% with a second test. Similarly, in smear-negative patients, sensitivity increased from 23.8% to 31.1%. In a non-high-tuberculosis-burden setting, performing a second sputum NAAT significantly improves diagnostic sensitivity and shortens the time to diagnosis.

**IMPORTANCE** The standard diagnostic protocol for pulmonary tuberculosis still relies on collecting three consecutive sputum specimens to maximize the diagnostic yield of smear and culture. While nucleic acid amplification tests (NAATs) have become increasingly important diagnostic tools, the incremental value of this approach for sputum NAAT-based diagnosis has remained uncertain. In a large cohort from a non-high-tuberculosis-burden setting, we show that adding a second sputum NAAT meaningfully improves cumulative sensitivity and shortens the mean time to diagnosis, whereas a third sputum NAAT provides only modest additional benefit. Notably, a two-sputum NAAT strategy optimizes yield irrespective of smear status, achieving saturation in smear-positive cases and securing a modest but steady stepwise gain in sensitivity for smear-negative patients. Therefore, our findings provide crucial evidence to support a two-sputum NAAT strategy to optimize the diagnostic workflow for PTB in non-high-tuberculosis-burden settings.

**KEYWORDS** *Mycobacterium tuberculosis*, TB-PCR, NAAT, three consecutive sputum specimens

Pulmonary tuberculosis (PTB) remains a major global public health challenge, accounting for approximately 10.8 million new cases and 1.25 million deaths annually (1). The standard diagnostic approach for PTB is microbiological examination of sputum specimens. In patients with active PTB, the cumulative diagnostic

**Peer Reviewer** Harley T. Harris, University of Utah Health, Salt Lake City, Utah, USA

Address correspondence to Naohisa Urabe, naohisa.urabe@med.toho-u.ac.jp.

The authors declare no conflict of interest.

yield of induced sputum testing increases with each additional specimen. Reported yields for acid-fast bacillus (AFB) smear, nucleic acid amplification testing (NAAT), and culture are approximately 64%, 89%, and 70% for a single specimen, increasing to 91%, 99%, and 99% with three specimens, respectively (2). Accordingly, to maximize the diagnostic sensitivity of conventional microscopy and culture, the 2017 American Thoracic Society/Infectious Diseases Society of America/Centers for Disease Control and Prevention guidelines recommend the collection of three sputum specimens (3).

In clinical practice, however, achieving a definitive diagnosis in patients with suspected PTB often remains challenging, even after three sputum examinations, and invasive diagnostic procedures such as bronchoscopy are frequently required. Given the rapid turnaround time and high sensitivity of NAATs, implementing a multi-specimen NAAT strategy could improve diagnostic yield and potentially reduce reliance on such invasive procedures. Although the incremental diagnostic yield of performing consecutive NAATs has been explored several times, the existing evidence has some notable limitations. For example, a few of these studies incorporated a reference standard that required culture-confirmed tuberculosis (TB), and most were conducted either in high-burden settings or involved relatively small cohorts in low-burden countries. Consequently, the generalizability of these prior findings to non-high-burden settings remains uncertain.

Therefore, in this retrospective study, we sought to address this gap by evaluating the incremental diagnostic yield and time to diagnosis associated with up to three consecutive sputum NAATs in a non-high-burden country.

## MATERIALS AND METHODS

### Facility

This study was conducted at Toho University Omori Medical Center, a tertiary care and regional referral hospital in Tokyo, Japan. During the study period, Japan transitioned from an intermediate- to a low-TB-burden country, with the national incidence rate falling below 10 cases per 100,000 population in 2021 (4). In 2022, the incidence in the hospital's catchment area was 8.2 cases per 100,000 population, similar to the national average (4, 5). From January 2010 to March 2024, the institutional infection-control protocol required that patients with suspected PTB be placed under airborne precautions, which were maintained until three consecutive sputum specimens tested negative by both AFB smear and NAAT.

### Study population

Between 1 January 2010 and 31 December 2023, a total of 13,375 patients underwent sputum AFB smear testing at our center (Fig. 1). These patients were initially classified into two groups: a PTB group (n = 404) and a non-PTB group (n = 12,971).

The PTB group comprised patients who met at least one of the following criteria: (i) isolation of *Mycobacterium tuberculosis* complex by culture from any respiratory specimen (sputum, bronchoalveolar lavage fluid, lung tissue from biopsy or surgery), gastric aspirate, or pleural fluid (Table 1); or (ii) a positive NAAT result from a respiratory specimen together with clinical features and radiological findings consistent with active PTB. We excluded patients who (i) had been diagnosed with PTB at other hospitals (n = 14) or (ii) had received a clinical PTB diagnosis without microbiological confirmation using either culture or NAAT (n = 70). Patients who did not meet the PTB criteria were classified as non-PTB (n = 12,971).

Next, we applied specimen-based case-selection criteria. Initially, we excluded all patients who did not submit three consecutive sputum specimens tested by both sputum NAAT and sputum culture. After this step, 198 patients remained in the PTB group with 3,761 patients in the non-PTB group. However, of the 206 excluded PTB patients, 92 were re-included because their diagnosis had already been microbiologically

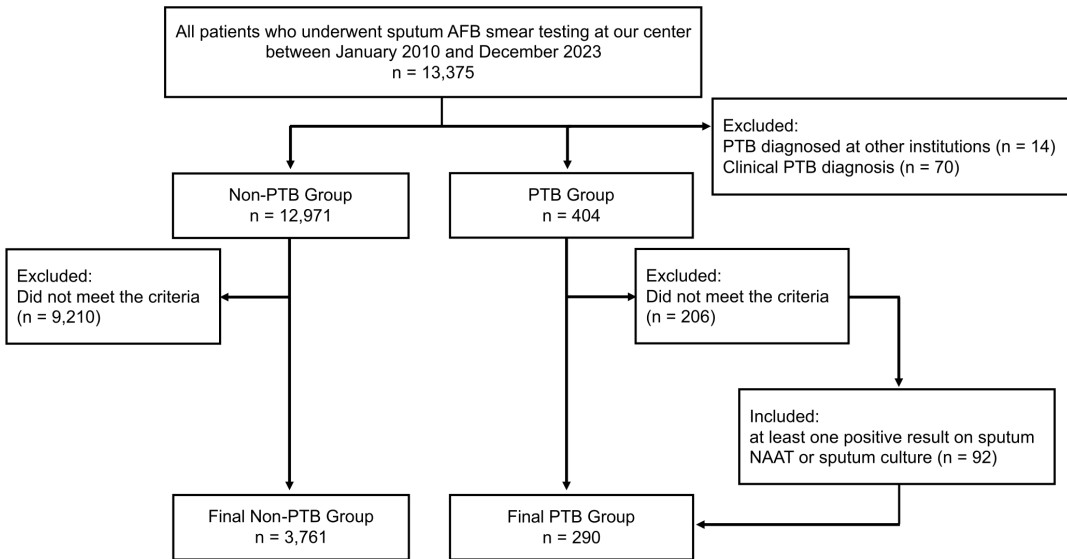

**FIG 1** Flow diagram of patient selection and classification.

confirmed by the first or second specimen, making further sampling unnecessary. Thus, the final study population comprised 4,051 patients: 290 in the PTB group and 3,761 in the non-PTB group.

## Study design

We conducted four primary analyses. First, we evaluated the cumulative sensitivity and specificity of sputum AFB smear, NAAT, and culture after one, two, and three consecutive sputum specimens. Second, we stratified the diagnostic yield by patient-level sputum smear status, comparing a smear-positive subgroup (*n* = 139; ≥1 positive AFB smear) with a smear-negative subgroup (*n* = 151; all three smears negative). Third, we performed a per-specimen analysis of sputum NAAT and culture positivity rates for all 834 sputum specimens, stratified by the corresponding smear result. Fourth, in patients with sputum-confirmed PTB, we compared the time from initial sputum collection to the first positive result for four diagnostic strategies (culture alone vs culture combined with one, two, or three NAATs).

In addition, we performed a sensitivity analysis of the cumulative diagnostic yield restricted to patients diagnosed exclusively from non-invasive specimens (sputum or gastric aspirate). This restriction was intended to mitigate verification bias introduced by patients diagnosed via invasive procedures, who often have paucibacillary, smear-negative disease, which could therefore lead to underestimation of the true sensitivity of sputum NAAT.

**TABLE 1** Patient characteristics and diagnostic specimen type

| Characteristic | PTB group (*n* = 290) | Non-PTB group (*n* = 3,761) |
|---|---|---|
| Sex: male: *n* (%) | 205 (70.7%) | 2,365 (62.9%) |
| Age: mean (SD) | 58.2 (20.4) | 68.5 (15.1) |
| Diagnostic specimen type, *n* (%) | | |
| Sputum | 201 (69.3%) | –[a] |
| Bronchoscopy | 61 (21.0%) | – |
| Gastric aspirate | 17 (5.9%) | – |
| Lung tissue | 6 (2.1%) | – |
| Pleural fluid | 5 (1.7%) | – |

[a]–, not applicable.

## Data collection

Clinical data were retrospectively extracted from the hospital's electronic health records for the study period. For all eligible patients, we collected demographic variables (age and sex), and the results of smear microscopy, NAAT, and mycobacterial culture performed on up to three consecutive sputum specimens. For patients in the PTB group, we additionally recorded: (i) the diagnostic modality confirming PTB (sputum, bronchoscopy specimen, gastric aspirate, lung tissue, or pleural fluid); and (ii) the dates of collection and reporting for each sputum specimen, to calculate the time from initial sputum collection to the first positive result for each diagnostic strategy.

## Microbiological examination

### Specimen processing

Sputum specimens were decontaminated and concentrated using the N-acetyl-L-cysteine-sodium hydroxide (NALC-NaOH) method. Briefly, 1 mL of sputum was mixed with an equal volume of NALC-NaOH solution and incubated for 15 min. The mixture was then diluted with phosphate-buffered saline (PBS) to a final volume of 15 mL and centrifuged at $3,000 \times g$ for 15 min. After discarding the supernatant, the pellet was resuspended in 1 mL of PBS. This processed specimen was used for smear microscopy, culture, and NAAT.

### Smear microscopy

Smears were prepared from both processed and unprocessed sputum specimens, stained using Ziehl-Neelsen and auramine fluorescence methods, and examined by light and fluorescence microscopy, respectively.

### NAAT for M. tuberculosis

NAAT was performed on the NALC-NaOH-processed specimens using the cobas TaqMan48 system with cobas MTB reagents (Roche Diagnostics, Basel, Switzerland), according to the manufacturer's instructions. This assay has demonstrated performance comparable to the Xpert MTB/RIF assay (Cepheid, Sunnyvale, CA, USA) in previous studies (6).

### Mycobacterial culture

A 0.1 mL aliquot of each decontaminated specimen was inoculated onto 2% Ogawa egg-based medium (Kyokuto Pharmaceutical Industrial Co., Tokyo, Japan) and incubated aerobically at 37°C for up to 8 weeks. Cultures were visually inspected weekly for colony growth. Mycobacterial colonies were identified as *M. tuberculosis* complex using the cobas MTB assay as described above or the Capilia TB assay (TAUNS, Shizuoka, Japan). Culture contamination was monitored with a prespecified target rate of ≤5% for sputum specimens processed by the NALC-NaOH method.

## Statistical analysis

Sensitivity and specificity were calculated with 95% confidence intervals (CI) using the Wilson score method. Time to diagnosis was defined as the interval from the initial sputum collection to the first positive result (by either NAAT or culture). This interval was summarized as mean ± standard deviation (SD) and median with interquartile range (IQR). In addition, the proportions of cases diagnosed within 7 and 14 days were calculated. All analyses were conducted using R version 4.3.2 (7).

## RESULTS

### Patient characteristics

A total of 4,051 patients were included in the analysis (Table 1). Patients in the PTB group were younger than those in the non-PTB group (mean age, 58.2 vs 68.5 years) and

**TABLE 2** Cumulative diagnostic yield of consecutive sputum testing[a]

| Test | No. of specimens | TP | FP | FN | TN | Sensitivity % (95% CI) | Specificity % (95% CI) |
|---|---|---|---|---|---|---|---|
| AFB smear | 1 | 96 | 115 | 194 | 3,646 | 33.1 (27.9–38.7) | 96.9 (96.3–97.4) |
| | 2 | 125 | 169 | 165 | 3,592 | 43.1 (37.5–48.9) | 95.5 (94.8–96.1) |
| | 3 | 139 | 192 | 151 | 3,569 | 47.9 (42.2–53.7) | 94.9 (94.2–95.6) |
| NAAT | 1 | 154 | 1 | 136 | 3,760 | 53.1 (47.4–58.8) | 99.97 (99.94–100) |
| | 2 | 183 | 1 | 107 | 3,760 | 63.1 (57.4–68.5) | 99.97 (99.94–100) |
| | 3 | 195 | 1 | 95 | 3,760 | 67.2 (61.6–72.4) | 99.97 (99.94–100) |
| Culture | 1 | 111 | 0 | 179 | 3,761 | 38.3 (32.9–43.9) | 100 (99.9–100) |
| | 2 | 154 | 0 | 136 | 3,761 | 53.1 (47.4–58.8) | 100 (99.9–100) |
| | 3 | 174 | 0 | 116 | 3,761 | 60.0 (54.3–65.5) | 100 (99.9–100) |

[a]TP, true positive; FP, false positive; FN, false negative; TN, true negative.

were more frequently male (70.7% vs 62.9%). In the PTB group ($n = 290$), the diagnosis was most commonly established from sputum specimens ($n = 201$, 69.3%), followed by bronchoscopy specimens ($n = 61$, 21.0%), gastric aspirates ($n = 17$, 5.9%), lung tissue ($n = 6$, 2.1%), and pleural fluid ($n = 5$, 1.7%).

## Cumulative diagnostic yield of consecutive sputum testing

In the overall PTB group ($n = 290$), the cumulative sensitivity of sputum AFB smear, NAAT, and culture increased with each additional sputum specimen (Table 2). For AFB smear, the cumulative sensitivities (95% CI) after one, two, and three specimens were 33.1% (27.9%–38.7%), 43.1% (37.5%–48.9%), and 47.9% (42.2%–53.7%), respectively. For NAAT, the corresponding sensitivities were 53.1% (47.4%–58.8%), 63.1% (57.4%–68.5%), and 67.2% (61.6%–72.4%). For culture, they were 38.3% (32.9%–43.9%), 53.1% (47.4%–58.8%), and 60.0% (54.3%–65.5%), respectively. The specificity of both NAAT and culture remained consistently above 99% regardless of the number of specimens.

## Cumulative diagnostic yield stratified by sputum smear status

Cumulative diagnostic yield was further evaluated after stratification by patient-level sputum smear status (Table 3). In the smear-positive PTB group ($n = 139$), NAAT cumulative sensitivities (95% CI) for one, two, and three specimens were 84.9% (78.0%–89.9%), 97.8% (94.0%–99.2%), and 99.3% (95.8%–99.9%), respectively. Culture sensitivities were 50.4% (42.2%–58.5%), 70.5% (62.6%–77.3%), and 76.3% (68.7%–82.6%). In the smear-negative PTB group ($n = 151$), NAAT sensitivities were 23.8% (17.6%–31.4%), 31.1% (24.2%–38.9%), and 37.7% (30.4%–45.7%). Culture sensitivities were 27.2% (20.6%–34.9%), 37.1% (29.6%–45.4%), and 45.0% (37.3%–53.0%). Specificity for both NAAT and culture remained consistently above 99% in both subgroups, irrespective of the number of specimens.

## Per-specimen analysis of NAAT and culture

A per-specimen analysis was conducted on 834 sputum specimens (Table 4). Overall, 246 (29.5%) specimens were positive by both NAAT and culture, 141 (16.9%) were NAAT-positive but culture-negative, 62 (7.4%) were NAAT-negative but culture-positive, and 385 (46.2%) were negative by both methods. The corresponding per-specimen positivity rates were 387/834 (46.4%) for NAAT and 308/834 (36.9%) for culture.

Among 244 smear-positive specimens, 166 (68.0%) were positive by both NAAT and culture, 73 (29.9%) were NAAT-positive but culture-negative, 1 (0.4%) was NAAT-negative but culture-positive, and 4 (1.6%) were negative by both tests. In this subgroup, the per-specimen positivity rates were 239/244 (98.0%) for NAAT and 167/244 (68.4%) for culture.

Among 590 smear-negative specimens, 80 (13.6%) were positive by both methods, 68 (11.5%) were NAAT-positive but culture-negative, 61 (10.3%) were NAAT-negative but culture-positive, and 381 (64.6%) were negative by both tests. In smear-negative

**TABLE 3** Cumulative diagnostic yield stratified by sputum smear status[a]

| Test | No. of specimens | TP | FP | FN | TN | Sensitivity % (95% CI) | Specificity % (95% CI) |
|---|---|---|---|---|---|---|---|
| Smear-positive PTB group (n = 139) | | | | | | | |
| NAAT | 1 | 118 | 0 | 21 | 192 | 84.9 (78.0–89.9) | 100 (98.04–100) |
| | 2 | 136 | 0 | 3 | 192 | 97.8 (94.0–99.2) | 100 (98.04–100) |
| | 3 | 138 | 0 | 1 | 192 | 99.3 (95.8–99.9) | 100 (98.04–100) |
| Culture | 1 | 70 | 0 | 69 | 192 | 50.4 (42.2–58.5) | 100 (98.04–100) |
| | 2 | 98 | 0 | 41 | 192 | 70.5 (62.6–77.3) | 100 (98.04–100) |
| | 3 | 106 | 0 | 33 | 192 | 76.3 (68.7–82.6) | 100 (98.04–100) |
| Smear-negative PTB group (n = 151) | | | | | | | |
| NAAT | 1 | 36 | 1 | 115 | 3,569 | 23.8 (17.6–31.4) | 99.97 (99.82–100) |
| | 2 | 47 | 1 | 104 | 3,569 | 31.1 (24.2–38.9) | 99.97 (99.82–100) |
| | 3 | 57 | 1 | 94 | 3,569 | 37.7 (30.4–45.7) | 99.97 (99.82–100) |
| Culture | 1 | 41 | 0 | 110 | 3,569 | 27.2 (20.6–34.9) | 100 (99.88–100) |
| | 2 | 56 | 0 | 95 | 3,569 | 37.1 (29.6–45.4) | 100 (99.88–100) |
| | 3 | 68 | 0 | 83 | 3,569 | 45.0 (37.3–53.0) | 100 (99.88–100) |

[a]TP, true positive; FP, false positive; FN, false negative; TN, true negative.

specimens, the per-specimen positivity rates were 148/590 (25.1%) for NAAT and 141/590 (23.9%) for culture.

## Time to diagnosis

Among patients with sputum-confirmed PTB, the time from initial sputum collection to the first positive microbiological result was compared across diagnostic strategies (Table 5). With culture alone, the mean time to diagnosis was 27.7 ± 12.6 days (median, 24 days; IQR, 21–32); no patients were diagnosed within 7 days, and only 5.7% were diagnosed within 14 days. By contrast, a single-NAAT strategy reduced the mean time to diagnosis to 10.9 ± 16.3 days (median, 3 days; IQR, 3–3), with 75.5% of patients diagnosed within 7 days. A two-NAAT strategy further shortened the mean diagnostic time to 8.3 ± 13.0 days (median, 3 days; IQR, 3–3), and 84.8% were diagnosed within 7 days. With three NAATs, the mean time to diagnosis was 8.1 ± 13.2 days (median, 3 days; IQR, 3–3), and 86.1% of patients were diagnosed within 7 days.

## Cumulative diagnostic yield in the non-invasive subgroup

In a sensitivity analysis restricted to 218 patients diagnosed exclusively via non-invasive specimens (sputum [n = 201] or gastric aspirate [n = 17]; Table 6), the cumulative sensitivities (95% CI) after one, two, and three sputum specimens were: 43.1% (36.7%–49.8%), 54.6% (48.0%–61.1%), and 60.1% (53.5%–66.4%) for AFB smear; 68.4% (61.9%–74.2%), 79.8% (74.0%–84.6%), and 83.0% (77.5%–87.4%) for NAAT; and 46.3% (39.8%–53.0%), 64.2% (57.7%–70.3%), and 72.0% (65.7%–77.6%) for culture, respectively.

**TABLE 4** Per-specimen NAAT and culture positivity analysis

| Group | NAAT + / culture + (%) | NAAT + / culture − (%) | NAAT − / culture + (%) | NAAT − / culture − (%) | Total NAAT positive (%) | Total culture positive (%) |
|---|---|---|---|---|---|---|
| Overall (n = 834) | 246 (29.5%) | 141 (16.9%) | 62 (7.4%) | 385 (46.2%) | 387 (46.4%) | 308 (36.9%) |
| Smear-positive (n = 244) | 166 (68.0%) | 73 (29.9%) | 1 (0.4%) | 4 (1.6%) | 239 (98.0%) | 167 (68.4%) |
| Smear-negative (n = 590) | 80 (13.6%) | 68 (11.5%) | 61 (10.3%) | 381 (64.6%) | 148 (25.1%) | 141 (23.9%) |

**TABLE 5** Time to diagnosis based on different diagnostic strategies

| Method | N | Mean ± SD (days) | Median (IQR) (days) | ≤7 days (%) | ≤14 days (%) |
|---|---|---|---|---|---|
| Culture only | 157 | 27.7 ± 12.6 | 24.0 (21.0–32.0) | 0 (0.0 %) | 9 (5.7 %) |
| Culture + 1 NAAT | 192 | 10.9 ± 16.3 | 3.0 (3.0–3.0) | 145 (75.5 %) | 145 (75.5 %) |
| Culture + 2 NAATs | 198 | 8.3 ± 13.0 | 3.0 (3.0–3.0) | 168 (84.8 %) | 168 (84.8 %) |
| Culture + 3 NAATs | 201 | 8.1 ± 13.2 | 3.0 (3.0–3.0) | 173 (86.1 %) | 173 (86.1 %) |

## DISCUSSION

This study assessed the diagnostic performance of up to three consecutive sputum NAATs in patients with suspected PTB in a setting transitioning to a low TB burden. Our results demonstrate that the second NAAT provides a substantial incremental yield, increasing cumulative sensitivity by 10.0 percentage points (from 53.1% to 63.1%) and shortening the mean time to diagnosis by 2.6 days (from 10.9 ± 16.3 to 8.3 ± 13.0 days). In contrast, the third NAAT yielded a marginal gain, increasing sensitivity by 4.1 percentage points (to 67.2%) and shortening the diagnostic time by 0.2 days (to 8.1 ± 13.2 days).

Our study has several strengths that enhance its contribution to the field. It was conducted using a large cohort (290 PTB and 3,761 non-PTB) in Japan, a country that has recently transitioned to a low-burden status. Unlike most prior studies, our reference standard for confirmed PTB incorporated findings from invasive procedures such as bronchoscopy. A key advantage of this rigorous approach is the inclusion of 151 smear-negative PTB patients, a diagnostically challenging subgroup represented by a considerably larger sample size than in most prior studies from low-burden settings. Although previous investigations have examined the incremental diagnostic yield of consecutive sputum NAATs in both high- and low-burden settings (8–11), these studies were generally limited by smaller sample sizes or reported minimal incremental gains when baseline sensitivity was already high. By evaluating a sizeable cohort that includes a substantial smear-negative population in a non-high-burden context, our findings strengthen the evidence and improve the generalizability of consecutive sputum NAAT strategies to routine clinical practice in similar settings.

The diagnostic utility of consecutive sputum NAATs is strongly dependent on sputum smear status. In smear-positive PTB, cumulative NAAT sensitivity was already high with the first specimen (84.9%) and approached saturation with additional specimens (97.8% with two and 99.3% with three), indicating that up to two NAATs are sufficient for diagnosis in most smear-positive cases. In smear-negative PTB, by contrast, cumulative NAAT sensitivity increased more modestly but steadily with each additional specimen (23.8%, 31.1%, and 37.7% for one, two, and three specimens, respectively). Although three cultures achieved a higher cumulative sensitivity in this subgroup (45.0%), the incremental gains and substantially shorter turnaround time of consecutive sputum NAATs offer important clinical advantages, particularly for early treatment and infection-control decisions. This is particularly important because smear-negative disease,

**TABLE 6** Cumulative diagnostic yield in the non-invasive subgroup[a]

| Test | No. of specimens | TP | FP | FN | TN | Sensitivity % (95% CI) | Specificity % (95% CI) |
|---|---|---|---|---|---|---|---|
| AFB smear | 1 | 94 | 115 | 124 | 3,646 | 43.1 (36.7–49.8) | 96.9 (96.3–97.4) |
| | 2 | 119 | 169 | 99 | 3,592 | 54.6 (48.0–61.1) | 95.5 (94.8–96.1) |
| | 3 | 131 | 192 | 87 | 3,569 | 60.1 (53.5–66.4) | 94.9 (94.2–95.6) |
| NAAT | 1 | 149 | 1 | 69 | 3,760 | 68.4 (61.9–74.2) | 99.97 (99.94–100) |
| | 2 | 174 | 1 | 44 | 3,760 | 79.8 (74.0–84.6) | 99.97 (99.94–100) |
| | 3 | 181 | 1 | 37 | 3,760 | 83.0 (77.5–87.4) | 99.97 (99.94–100) |
| Culture | 1 | 101 | 0 | 117 | 3,761 | 46.3 (39.8–53.0) | 100 (99.9–100) |
| | 2 | 140 | 0 | 78 | 3,761 | 64.2 (57.7–70.3) | 100 (99.9–100) |
| | 3 | 157 | 0 | 61 | 3,761 | 72.0 (65.7–77.6) | 100 (99.9–100) |

[a]TP, true positive; FP, false positive; FN, false negative; TN, true negative.

once considered less transmissible, is now recognized as a significant contributor to TB transmission (12), making expedited diagnosis a priority. Therefore, performing consecutive sputum NAATs represents a promising strategy for expediting diagnosis in this patient population.

In addition to increasing sensitivity, consecutive sputum NAATs provide other clinical benefits. NAAT can detect *M. tuberculosis* DNA even in culture-negative PTB, which accounts for approximately 15% of all TB cases (13). In our study, 16.9% of all sputum specimens were NAAT-positive but culture-negative. Furthermore, because prior fluoroquinolone exposure can delay culture positivity (14), consecutive sputum NAATs may offer a more robust diagnostic yield in such patients. Taken together, these findings suggest that maximizing the diagnostic yield with consecutive sputum NAATs on non-invasively collected sputum can reduce reliance on invasive procedures such as bronchoscopy. Minimizing the need for such interventions is clinically critical, given the procedural risks and patient burden associated with bronchoscopy, particularly in vulnerable populations including children and older adults.

Collectively, these findings support a pragmatic diagnostic strategy in which sputum NAAT is performed twice for most patients with suspected PTB. A two-sputum NAAT strategy captured the majority of PTB cases, substantially increased the cumulative yield compared with a single test, and markedly shortened the time to diagnosis. However, the incremental benefit of a third NAAT was modest for both absolute sensitivity and diagnostic timelines. Importantly, similar patterns were observed in the non-invasive subgroup, where cumulative sputum NAAT sensitivity increased from 68.4% with one specimen to 79.8% with two, and only marginally further to 83.0% with three. These results underscore the fact that among patients diagnosable using non-invasive specimens, the major incremental benefit of consecutive sputum NAAT is realized before the need arises to escalate to bronchoscopy or surgical sampling.

This study has several limitations. First, it was a single-center retrospective observational study, and the clinical decision to obtain three sputum specimens or proceed directly to invasive diagnostics was at the discretion of the treating physician, which may have introduced selection bias. Second, we lacked data on important variables such as specimen quality, use of induced sputum, and collection timing (e.g., early morning vs spot sputum); these unmeasured factors could have influenced test performance. Third, mycobacterial culture was performed exclusively on Ogawa solid medium; liquid culture systems, which are faster and more sensitive, were not employed. This may have resulted in an underestimation of culture sensitivity and delayed microbiological confirmation (15). Fourth, our analysis was limited to the cobas MTB assay; newer, potentially more sensitive NAATs, such as Xpert MTB/RIF Ultra, may demonstrate a different performance profile. Last, cost-effectiveness—an important consideration in resource-limited settings—was not evaluated. Despite these limitations, our study provides important evidence on the utility of consecutive sputum NAATs in a non-high-burden setting. Further studies are warranted to assess cost-effectiveness and to refine testing strategies across diverse epidemiologic and resource contexts.

## Conclusion

In a non-high-tuberculosis-burden setting, this study demonstrates that performing two consecutive sputum NAATs improves cumulative diagnostic sensitivity and shortens the time to diagnosis.

## ACKNOWLEDGMENTS

The authors thank Dr. Florence Ene (MBBS, PhD), Platinum Medical Consulting Japan, for professional medical English editing support.

Takumi Kanokogi and Naohisa Urabe were responsible for conceptualization, formal analysis, and writing the original draft. Hinako Murakami and Masakazu Sasaki provided resources. Data curation was performed by Takumi Kanokogi and Nozomi Tokita. Susumu

Sakamoto and Kazuma Kishi contributed to writing—review and editing. All authors reviewed and approved the final manuscript.

## AUTHOR AFFILIATIONS

[1]Department of Respiratory Medicine, Toho University Omori Medical Center, Tokyo, Japan

[2]Department of Respiratory Medicine, Toho University Graduate School of Medicine, Tokyo, Japan

[3]Department of Clinical Laboratory, Toho University Omori Medical Center, Tokyo, Japan

## AUTHOR ORCIDs

Takumi Kanokogi (ID) http://orcid.org/0000-0001-5901-232X
Naohisa Urabe (ID) http://orcid.org/0000-0003-4018-5364

## AUTHOR CONTRIBUTIONS

Takumi Kanokogi, Conceptualization, Data curation, Formal analysis, Writing – original draft | Naohisa Urabe, Conceptualization, Formal analysis, Writing – original draft | Nozomi Tokita, Data curation | Hinako Murakami, Resources | Masakazu Sasaki, Resources | Susumu Sakamoto, Writing – review and editing | Kazuma Kishi, Writing – review and editing

## ETHICS APPROVAL

This study was approved by the Ethics Committee of Toho University Omori Medical Center (Approval No. M24117). The requirement for individual informed consent was waived due to the retrospective nature of the study and use of the anonymized data. Study details were disclosed on the hospital website, affording patients the opportunity to opt out.

## ADDITIONAL FILES

The following material is available online.

Open Peer Review

**PEER REVIEW HISTORY (review-history.pdf).** An accounting of the reviewer comments and feedback.

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
