## [Reviewer comments · Microbiology Spectrum]

Microbiology Spectrum

Incremental Diagnostic Yield of Consecutive Sputum Nucleic Acid Amplification Tests for Pulmonary Tuberculosis

Takumi Kanokogi, Naohisa Urabe, Nozomi Tokita, Hinako Murakami, Masakazu Sasaki, Susumu Sakamoto, and Kazuma Kishi

Corresponding Author(s): Naohisa Urabe, Toho Daigaku

Review Timeline:

Submission Date:	August 18, 2025
Editorial Decision:	October 3, 2025
Revision Received:	December 1, 2025
Accepted:	December 19, 2025

Editor: Salika Shakir

Reviewer(s): Disclosure of reviewer identity is with reference to reviewer comments included in decision letter(s). The following individuals involved in review of your submission have agreed to reveal their identity: Harley T Harris (Reviewer #2)

Transaction Report:

DOI: <https://doi.org/10.1128/spectrum.02442-25>

Re: Spectrum02442-25 (Incremental Diagnostic Yield of Consecutive Sputum Nucleic Acid Amplification Tests for Pulmonary Tuberculosis)

Dear Dr. Naohisa Urabe:

Thank you for the privilege of reviewing your work. Your manuscript was reviewed by me and experts in the field. Below you will find my comments, instructions from the Spectrum editorial office, and the reviewer comments.

There were concerns that the English language usage in the manuscript might make it difficult to properly evaluate the science. The ASM Journals webpage provides links to various language editing services (<https://journals.asm.org/writing-your-paper#language-editing-services>). You may consider using these services when revising your manuscript. The use of these services will have no direct bearing on the editorial decision. ASM has no affiliation with these companies.

Revision Guidelines

Sincerely,
Salika Shakir
Editor
Microbiology Spectrum

Reviewer #1 (Comments for the Author):

The authors present a timely and relevant study about the effects of NAAT retesting patients suspected of pulmonary MTB

infection, particularly in a non-high burden setting. This is an important study to understand the diagnostic yield, if any, of performing multiple MTB NAAT tests as this practice can impact treatment and isolation precautions of the patient. Overall, this is a high-quality study that will advance the field. I respectfully ask the authors to address the following questions/comments:

Line 93

PTB+ group inclusion by culture can be from many respiratory specimens, but for NAAT, is sputum the only specimen type? Also is this raw and/or processed sputum? This is a bit confusing since the title of the manuscript specifically calls out sputum. There are different performance characteristics based on specimen type and only some of which is FDA approved.

Line 68

"not well established" is unfair since Cowan and others have published similar studies. Please be specific about which aspects of this study are novel and important to highlight. I believe this study has many.

Line 136

Please speak to the target bacterial/fungal contamination rate of this method of AFB culture.

Results:

Although not necessary, I recommend authors stratify based on if the patient received a bronchoscopy or not. As the authors states, this is an invasive procedure. The decision for this procedure to happen may skew the severity of symptoms and case complexity. Additionally, smear, culture, and NAAT all have higher sensitivities from BAL compared to sputum.

Reviewer #2 (Comments for the Author):

Major Comments for Authors:

- Throughout the text and table titles authors use "sputum specimens", "sputum samples", "sputum NAATs", etc. where it is not always referring to only sputum specimen analysis. This creates some confusion for the reader regarding whether the 89 samples from non-sputum sources are included or not. Authors should review the manuscript text and omit using the word sputum when not referring to the sputum-only analysis section or handling of sputum in methods. Some important example points for this include line 37, 102, 108, 118-119 (specifically indicate which specimen types did not undergo decontamination as applicable), figure 1, and table 2 title. Alternatively, if the 89 non-sputum samples were excluded from all analysis sets, then the information of other specimen types is not needed and can be removed from text and table 1. Either way, this needs to be addressed to allow for readers to properly understand the data presented.
- In the discussion, the authors reference other similar studies that have been done previously on the matter. The way this is presented significantly weakens the statement of importance of this paper and its novelty: being performed on a larger population and in a moderate to low TB burden country. While this is pointed out eventually, it is after 1.5 paragraphs of discussion of other studies- a section that does not add much to the paper either. The discussion section should focus primarily on your findings and your study. The authors should briefly summarize the other studies done before in context with how their study is different/ adds to the field/ improves. Authors can note other studies have found similar results but were conducted on smaller cohorts and in areas of differing TB burden than this study, without needing to list all the details of other published studies.

Minor Comments for Authors:

- Line 34-35: Suggest inclusion that sensitivity increased for smear positives with addition of second NAAT.
- Line 37-38/ Line 268-272: The abstract conclusion is different than the text discussion section. The abstract proposes 2 NAATs for all but the discussion indicates this is for smear-negative patients. Adjust these sections to be in concordance with each other.
- Line 54/ Reference 1: The WHO 2024 TB report is available, and this number of infections and reference can be updated.
- Line 91-92 and Fig 1: There is conflicting information on whether patients were included or excluded if they did not meet the 3 specimen criteria. The methods section indicates they were included in analysis but figure 1 shows these to be excluded. Please correct accordingly to reflect what was done or indicate if they were initially included but then excluded from final analysis group.
- Line 94: CLSI separates pleural fluid and gastric aspirate as separate specimen types from respiratory. Change text to respiratory sources or just specimen. Also add in reference to Table 1 for specimen type breakdown.
- Line 168-172 and 179-186: This data presentation is bulky and difficult to read through. Some suggestions are to break up into multiple sentences by test method. Also, you do not need to include the 95% CI every time. Include as "The cumulative sensitivities (95% CI) for..." which will reduce the sentence length down.
- Line 279-280: Authors could also note that liquid culture takes less time than solid media growth and may increase the time to diagnosis.
- Figure 1: please clarify either here or in text what a clinical diagnosis of PTB means for those excluded. Was this previous diagnosis, diagnosis without microbiologic testing, or something else?
- Table 1: suggest changing "final diagnostic method" to "diagnostic specimen type". Method is generally more synonymous with testing modality than specimen type.
- Table 2 and Table 3: suggest adding in dividing lines between the test methods data presentation (ie. Line across between NAAT 3 row and Culture 1 row) to allow for better visualization by readers.

Journal Name: Microbiology Spectrum

Manuscript Number: Spectrum02442-25

Manuscript Title: Incremental Diagnostic Yield of Consecutive Sputum Nucleic Acid Amplification Tests for Pulmonary Tuberculosis

Overall Impressions and comments to editor:

This study analyzed the diagnostic yield of additional nucleic acid amplification tests (NAATs) for diagnosis of tuberculosis (TB) in a setting of moderate to low TB burden. Authors did a retrospective review of all patients tested over 13 years and determined how each successive NAAT test affected the diagnostic yield. Ultimately, the authors conclude that, particularly in smear-negative individuals, 2 NAAT tests gave significant increase in diagnostic yield over a single NAAT and 3 NAATS did not increase significantly from 2. This paper presents a new look at the utility of diagnostic methods in a moderate to low TB burden country, which has not been studied before. While the basis of the study is good, the paper execution has some flaws that hinder the reader's comprehension and understanding. The most notable is the usage of sputum specimens throughout the paper does not consistently match the data descriptions indicating non-sputum specimens were also included in some analyses but not others. It is difficult as a reader to fully understand when data is looking at sputum only or including the other respiratory sources. This is the biggest area needed addressing prior to publication as it is throughout the entire manuscript. Once this and other reviewer comments are addressed, the paper will be significantly improved to be considered for publication.

Major Comments for Authors:

- Throughout the text and table titles authors use "sputum specimens", "sputum samples", "sputum NAATs", etc. where it is not always referring to only sputum specimen analysis. This creates some confusion for the reader regarding whether the 89 samples from non-sputum sources are included or not. Authors should review the manuscript text and omit using the word sputum when not referring to the sputum-only analysis section or handling of sputum in methods. Some important example points for this include line 37, 102, 108, 118-119 (specifically indicate which specimen types did not undergo decontamination as applicable), figure 1, and table 2 title. Alternatively, if the 89 non-sputum samples were excluded from all analysis sets, then the information of other specimen types is not needed and can be removed from text and table 1. Either way, this needs to be addressed to allow for readers to properly understand the data presented.

- In the discussion, the authors reference other similar studies that have been done previously on the matter. The way this is presented significantly weakens the statement of importance of this paper and its novelty: being performed on a larger population and in a moderate to low TB burden country. While this is pointed out eventually, it is after 1.5 paragraphs of discussion of other studies- a section that does not add much to the paper either. The discussion section should focus primarily on your findings and your study. The authors should briefly summarize the other studies done before in context with how their study is different/ adds to the field/ improves. Authors can note other studies have found similar results but were conducted on smaller cohorts and in areas of differing TB burden than this study, without needing to list all the details of other published studies.

Minor Comments for Authors:

- Line 34-35: Suggest inclusion that sensitivity increased for smear positives with addition of second NAAT.
- Line 37-38/ Line 268-272: The abstract conclusion is different than the text discussion section. The abstract proposes 2 NAATs for all but the discussion indicates this is for smear-negative patients. Adjust these sections to be in concordance with each other.
- Line 54/ Reference 1: The WHO 2024 TB report is available, and this number of infections and reference can be updated.
- Line 91-92 and Fig 1: There is conflicting information on whether patients were included or excluded if they did not meet the 3 specimen criteria. The methods section indicates they were included in analysis but figure 1 shows these to be excluded. Please correct accordingly to reflect what was done or indicate if they were initially included but then excluded from final analysis group.
- Line 94: CLSI separates pleural fluid and gastric aspirate as separate specimen types from respiratory. Change text to respiratory sources or just specimen. Also add in reference to Table 1 for specimen type breakdown.
- Line 168-172 and 179-186: This data presentation is bulky and difficult to read through. Some suggestions are to break up into multiple sentences by test method. Also, you do not need to include the 95% CI every time. Include as “The cumulative sensitivities (95% CI) for...” which will reduce the sentence length down.
- Line 279-280: Authors could also note that liquid culture takes less time than solid media growth and may increase the time to diagnosis.
- Figure 1: please clarify either here or in text what a clinical diagnosis of PTB means for those excluded. Was this previous diagnosis, diagnosis without microbiologic testing, or something else?

- Table 1: suggest changing “final diagnostic method” to “diagnostic specimen type”. Method is generally more synonymous with testing modality than specimen type.
- Table 2 and Table 3: suggest adding in dividing lines between the test methods data presentation (ie. Line across between NAAT 3 row and Culture 1 row) to allow for better visualization by readers.

RESPONSES TO REVIEWERS' COMMENTS

We appreciate the reviewers' rigorous and comprehensive evaluation of our manuscript. In response, we have carried out a careful, point-by-point revision, incorporating all requested changes into the updated text. To our knowledge, these revisions have enhanced the presentation of this study. We respectfully submit the revised manuscript for further consideration and publication in *Journal of Clinical Microbiology*.

Reviewer 1

Comment 1: *Line 93: PTB+ group inclusion by culture can be from many respiratory specimens, but for NAAT, is sputum the only specimen type? Also is this raw and/or processed sputum? This is a bit confusion since the title of the manuscript specifically calls out sputum. There are different performance characteristics based on specimen type and only some of which is FDA approved.*

Response: Thank you for the comprehensive review. We especially appreciate this important comment regarding the need to clarify specimen types given the sputum-focused title. Regarding the specific question on processing, all sputum NAATs in this study were performed on processed specimens (NALC-NaOH pellets), as detailed in the "Specimen processing" section.

In our study, PTB cases were identified using culture results from a range of respiratory specimens (sputum, bronchoalveolar lavage fluid, lung tissue, gastric aspirate, and pleural fluid) as well as NAAT results from respiratory specimens, reflecting real-world diagnostic practice. However, all diagnostic yield analyses in this manuscript are restricted exclusively to sputum NAAT and sputum culture. No non-sputum NAAT results were included in the performance evaluation.

To make this distinction explicit and unambiguous, we have substantially revised the Methods section under Study Population to clarify 1) the microbiologic criteria used for PTB case definition and 2) the sputum-specific criteria used for cohort selection.

Revisions in the manuscript:

- *Methods (Page 5, Lines 90–108)*

“Study population

Between January 1, 2010, and December 31, 2023, a total of 13,375 patients underwent sputum AFB smear testing at our center [Fig 1]. These patients were initially classified into two groups: a PTB group (n = 404) and a non-PTB group (n = 12,971).

The PTB group comprised patients who met at least one of the following criteria: (1) isolation of Mycobacterium tuberculosis complex by culture from any respiratory specimen (sputum, bronchoalveolar lavage fluid, lung tissue from biopsy or surgery), gastric aspirate, or pleural fluid [Table 1]; or (2) a positive NAAT result from a respiratory specimen together with clinical features and radiologic findings consistent with active PTB. We excluded patients who (i) had been diagnosed with PTB at other hospitals (n = 14) or (ii) had received a clinical PTB diagnosis without microbiologic confirmation using either culture or NAAT (n = 70). Patients who did not meet the PTB criteria were classified as non-PTB (n = 12,971).

Next, we applied specimen-based case-selection criteria. Initially, we excluded all patients who did not submit three consecutive sputum specimens tested by both sputum NAAT and sputum culture. After this step, 198 patients remained in the PTB group with 3,761 patients in the non-PTB group. However, of the 206 excluded PTB patients, 92 were re-included because their diagnosis had already been microbiologically confirmed by the first or second specimen, making further sampling unnecessary. Thus, the final study population comprised 4,051 patients: 290 in the PTB group and 3,761 in the non-PTB group.”

- *Methods (Page 8, Lines 150–153)*

“NAAT for Mycobacterium tuberculosis

NAAT was performed on the NALC-NaOH-processed specimens using the cobas® TaqMan48 system with cobas MTB reagents (Roche Diagnostics, Basel, Switzerland), according to the manufacturer's instructions.”

Comment 2: *Line 68: "not well established" is unfair since Cowan and others have published similar studies. Please be specific about which aspects of this study are novel and important to highlight. I believe this study has many.*

Response: We agree with this assessment and appreciate the opportunity to clarify our

study's contribution. In response, we have removed the phrase “not well established” and revised the Introduction to: (1) explicitly acknowledge existing evidence on the incremental yield of consecutive NAATs, including pivotal studies such as those by Cowan et al.; and (2) clearly articulate the specific gaps that our study addresses. Specifically, we have emphasized the novelty of evaluating a large cohort in a setting transitioning to a low TB burden, utilizing a robust reference standard that incorporates invasive diagnostic procedures.

Revisions in the manuscript:

- *Introduction (Page 4, Lines 67–77)*

“Given the rapid turnaround time and high sensitivity of NAATs, implementing a multi-specimen NAAT strategy could improve diagnostic yield and potentially reduce reliance on such invasive procedures. Although the incremental diagnostic yield of performing consecutive NAATs has been explored severally, the existing evidence has some notable limitations. For example, few of these studies incorporated a reference standard that required culture-confirmed TB, and most were conducted either in high-burden settings or involved relatively small cohorts in low-burden countries. Consequently, the generalizability of these prior findings to non-high-burden settings remains uncertain.

Therefore, in this retrospective study, we sought to address this gap by evaluating the incremental diagnostic yield and time to diagnosis associated with up to three consecutive sputum NAATs in a non-high-burden country.”

Comment 3: *Line 136: Please speak to the target bacterial/fungal contamination rate of this method of AFB culture.*

Response: We appreciate this inquiry regarding the quality control benchmarks for our culture methods. In our laboratory, the target contamination rate for sputum specimens processed via the NALC–NaOH method is set at $\leq 5\%$, which is consistent with standard quality assurance guidelines. We have added this specific target to the Mycobacterial Culture subsection in the revised manuscript.

Revisions in the manuscript:

- *Methods (Page 8, Lines 161–163)*

“Culture contamination was monitored with a prespecified target rate of ≤5% for sputum specimens processed using the NALC–NaOH method.”

Comment 4: *Results: Although not necessary, I recommend authors stratify based on if the patient received a bronchoscopy or not. As the authors states, this is an invasive procedure. The decision for this procedure to happen may skew the severity of symptoms and case complexity. Additionally, smear, culture, and NAAT all have higher sensitivities from BAL compared to sputum.*

Response: The reviewer’s point regarding potential bronchoscopy-related selection bias is indeed valid. We agree that the decision to perform bronchoscopy likely reflects greater clinical severity and case complexity. Furthermore, because bronchoscopy is typically pursued when initial sputum-based testing fails to establish a diagnosis, patients in this subgroup often present with paucibacillary disease, which could skew the apparent sensitivity of sputum tests.

To address this and provide the recommended stratification, we conducted a sensitivity analysis restricted to patients whose PTB diagnosis was established exclusively by non-invasive specimens (sputum and/or gastric aspirate). This excludes patients for whom bronchoscopy or other invasive procedures were necessary for confirmation, thereby mitigating verification bias. The results of this analysis have been added to the Results section under "Cumulative Diagnostic Yield in the Non-Invasive Subgroup" and are presented in Supplementary Table 1. As noted in the Discussion, similar trends were observed in this subgroup, reinforcing the robustness of our main findings.

Revisions in the manuscript:

Results:

- *Methods (Page 6, Lines 120–124)*

“In addition, we performed a sensitivity analysis of the cumulative diagnostic yield restricted to patients diagnosed exclusively from non-invasive specimens (sputum or gastric aspirate). This restriction was intended to mitigate verification bias introduced by patients diagnosed via invasive procedures, who often have paucibacillary, smear-negative disease, which could therefore lead to underestimation of the true sensitivity of sputum NAAT.”

- *Results (Page 11, Lines 235–241)*

“Cumulative Diagnostic Yield in the Non-Invasive Subgroup

In a sensitivity analysis restricted to 218 patients diagnosed exclusively via non-invasive specimens (sputum, n = 201; gastric aspirate, n = 17) (Supplementary Table 1), the cumulative sensitivities (95% CI) after one, two, and three sputum specimens were 43.1% (36.7–49.8%), 54.6% (48.0–61.1%), and 60.1% (53.5–66.4%) for AFB smear; 68.4% (61.9–74.2%), 79.8% (74.0–84.6%), and 83.0% (77.5–87.4%) for NAAT; and 46.3% (39.8–53.0%), 64.2% (57.7–70.3%), and 72.0% (65.7–77.6%) for culture, respectively.”

Reviewer 2

Major comments

Comment 1: *Throughout the text and table titles authors use "sputum specimens", "sputum samples", "sputum NAATs", etc. where it is not always referring to only sputum specimen analysis. This creates some confusion for the reader regarding whether the 89 samples from non-sputum sources are included or not. Authors should review the manuscript text and omit using the word sputum when not referring to the sputum-only analysis section or handling of sputum in methods. Some important example points for this include line 37, 102, 108, 118-119 (specifically indicate which specimen types did not undergo decontamination as applicable), figure 1, and table 2 title. Alternatively, if the 89 non-sputum samples were excluded from all analysis sets, then the information of other specimen types is not needed and can be removed from text and table 1. Either way, this needs to be addressed to allow for readers to properly understand the data presented.*

Response: We sincerely thank the reviewer for this crucial observation. We apologize for the potential confusion caused by the ambiguous terminology regarding "sputum" versus "non-sputum" specimens. We agree that it is essential to clearly distinguish between specimens used for diagnostic performance analysis (strictly sputum) and those used for the reference standard/case definition (which included non-sputum sources). To address this, we have taken the following steps:

1. Clarification of Cohort vs. Analysis: As the reviewer noted, 89 PTB cases were

confirmed using non-sputum specimens (e.g., BAL fluid, gastric aspirate) because sputum testing was non-diagnostic or unavailable at the time of decision. We have clarified in the Methods (Study Population) that while these patients are included in the PTB group to ensure a robust reference standard, the diagnostic yield analyses (Tables 2–5) were performed exclusively on their sputum specimens. Non-sputum NAAT/culture results were strictly excluded from the performance calculations.

2. Systematic Text Revision: We have systematically reviewed the manuscript. We have removed "sputum" where the context refers to the general patient cohort or diagnosis, and retained/added it where the analysis is strictly limited to sputum.
3. Figure 1 and Table Titles: Figure 1 and Table titles have been updated to explicitly state "sputum" where the data refers strictly to sputum-based assays, ensuring readers can easily distinguish the analytical scope.

Revisions in the manuscript:

- *Methods (Page 5, Lines 90–108)*

“Study population

Between January 1, 2010, and December 31, 2023, a total of 13,375 patients underwent sputum AFB smear testing at our center [Fig 1]. These patients were initially classified into two groups: a PTB group (n = 404) and a non-PTB group (n = 12,971).

The PTB group comprised patients who met at least one of the following criteria: (1) isolation of Mycobacterium tuberculosis complex by culture from any respiratory specimen (sputum, bronchoalveolar lavage fluid, lung tissue from biopsy or surgery), gastric aspirate, or pleural fluid [Table 1]; or (2) a positive NAAT result from a respiratory specimen together with clinical features and radiologic findings consistent with active PTB. We excluded patients who (i) had been diagnosed with PTB at other hospitals (n = 14) or (ii) had received a clinical PTB diagnosis without microbiologic confirmation using either culture or NAAT (n = 70). Patients who did not meet the PTB criteria were classified as non-PTB (n = 12,971).

Next, we applied specimen-based case-selection criteria. Initially, we excluded all patients who did not submit three consecutive sputum specimens tested by both sputum NAAT and sputum culture. After this step, 198 patients remained in the PTB group with 3,761 patients in the non-PTB group. However, of the 206 excluded PTB patients, 92 were re-included because their diagnosis had already been

microbiologically confirmed by the first or second specimen, making further sampling unnecessary. Thus, the final study population comprised 4,051 patients: 290 in the PTB group and 3,761 in the non-PTB group.”

Comment 2: *In the discussion, the authors reference other similar studies that have been done previously on the matter. The way this is presented significantly weakens the statement of importance of this paper and its novelty: being performed on a larger population and in a moderate to low TB burden country. While this is pointed out eventually, it is after 1.5 paragraphs of discussion of other studies- a section that does not add much to the paper either. The discussion section should focus primarily on your findings and your study. The authors should briefly summarize the other studies done before in context with how their study is different/ adds to the field/ improves. Authors can note other studies have found similar results but were conducted on smaller cohorts and in areas of differing TB burden than this study, without needing to list all the details of other published studies.*

Response: We sincerely appreciate this constructive feedback regarding the structure and flow of the Discussion. We agree that the previous detailed recitation of prior studies distracted from the novelty and specific contributions of our own work. In accordance with your suggestion, we have significantly restructured this section. We have removed the study-by-study descriptions of prior work and replaced them with a concise synthesis. The revised text now prioritizes our study’s specific strengths—namely, the large sample size, the robust reference standard (incorporating invasive procedures), and the analysis of a substantial smear-negative cohort in a setting transitioning to a low TB burden. As reflected in the second paragraph of the revised Discussion, we now briefly reference earlier investigations only to contextualize how our study improves upon their limitations (e.g., small sample sizes or limited reference standards) and strengthens the generalizability of consecutive NAAT strategies.

Revisions in the manuscript:

- *Discussion (Page 12, Lines 251–263)*

“Our study has several strengths that enhance its contribution to the field. It was conducted using a large cohort (290 PTB and 3,761 non-PTB) in Japan, a country that

has recently transitioned to a low-burden status. Unlike most prior studies, our reference standard for confirmed PTB incorporated findings from invasive procedures such as bronchoscopy. A key advantage of this rigorous approach is the inclusion of 151 smear-negative PTB patients, a diagnostically challenging subgroup represented by a considerably larger sample size than in most prior studies from low-burden settings. Although previous investigations have examined the incremental diagnostic yield of consecutive sputum NAATs in both high- and low-burden settings [7-10], these studies were generally limited by smaller sample sizes or reported minimal incremental gains when baseline sensitivity was already high. By evaluating a sizeable cohort that includes a substantial smear-negative population in a non-high-burden context, our findings strengthen the evidence and improve the generalizability of consecutive sputum NAAT strategies to routine clinical practice in similar settings.”

Minor comments

Comment 1: *Line 34-35: Suggest inclusion that sensitivity increased for smear positives with addition of second NAAT.*

Response: We agree with this suggestion. The Abstract has been revised to specifically indicate that, among smear-positive PTB cases, NAAT sensitivity increased from 84.9% with a single test to 97.8% with a second test.

Revisions in the manuscript:

- *Abstract (Page 2, Lines 36–38)*

“In smear-positive PTB, a NAAT sensitivity increased from an already high 84.9% with a single test to 97.8% with a second test. Similarly, in smear-negative patients, sensitivity increased from 23.8% to 31.1%.”

Comment 2: *Line 37-38/ Line 268-272: The abstract conclusion is different than the text discussion section. The abstract proposes 2 NAATs for all but the discussion indicates this is for smear-negative patients. Adjust these sections to be in concordance with each other.*

Response: Thank you for highlighting this discrepancy. We agree that the messaging

should be consistent throughout the manuscript. Accordingly, we have revised the Discussion section to align with the Abstract. We now recommend a unified two-specimen NAAT strategy as the most pragmatic approach in clinical practice. While a single test detects most smear-positive PTB, a second test provides near-complete sensitivity in this group and significantly increases yield in smear-negative patients. Standardizing the recommendation to two NAATs simplifies the diagnostic algorithm, offering both consistency and practicability. The Discussion and Conclusion sections have been updated to reflect this unified recommendation.

Revisions in the manuscript:

- *Discussion (Page 13, Lines 288–297)*

“Collectively, these findings support a pragmatic diagnostic strategy in which sputum NAAT is performed twice for most patients with suspected PTB. A two-sputum NAAT strategy captured the majority of PTB cases, substantially increased the cumulative yield compared with a single test, and markedly shortened the time to diagnosis. However, the incremental benefit of a third NAAT was modest for both absolute sensitivity and diagnostic timelines. Importantly, similar patterns were observed in the non-invasive subgroup, where cumulative sputum NAAT sensitivity increased from 68.4% with one specimen to 79.8% with two, and only marginally further to 83.0% with three. These results underscore the fact that among patients diagnosable using non-invasive specimens, the major incremental benefit of consecutive sputum NAAT is realized before the need arises to escalate to bronchoscopy or surgical sampling.”

Comment 3: *Line 54/ Reference 1: The WHO 2024 TB report is available, and this number of infections and reference can be updated.*

Response: We appreciate the reviewer drawing attention to this update. The global PTB burden figures in the *Introduction* section have been updated to reflect the most recent estimates from the *2024 WHO Global Tuberculosis Report*. The text now reads, “approximately 10.8 million new cases and 1.25 million deaths annually.”

Page 4, Line 55. Reference 1 has been updated accordingly.

Comment 4: *Line 91-92 and Fig 1: There is conflicting information on whether*

patients were included or excluded if they did not meet the 3 specimen criteria. The methods section indicates they were included in analysis but figure 1 shows these to be excluded. Please correct accordingly to reflect what was done or indicate if they were initially included but then excluded from final analysis group.

Response: Thank you for pointing out this inconsistency.

To clarify, we have stated that patients who did not submit three consecutive sputum specimens were initially excluded. But those who had at least one positive sputum NAAT or sputum culture result were re-included in the PTB group, ensuring that all microbiologically confirmed PTB cases were captured in the final analysis. We have revised the *Study population* section and updated Figure 1 for consistency with the stepwise selection process used in our analysis.

Revisions in the manuscript:

- *Methods (Page 5, Lines 90–108)*

“Study population

Between January 1, 2010, and December 31, 2023, a total of 13,375 patients underwent sputum AFB smear testing at our center [Fig 1]. These patients were initially classified into two groups: a PTB group (n = 404) and a non-PTB group (n = 12,971).

The PTB group comprised patients who met at least one of the following criteria: (1) isolation of Mycobacterium tuberculosis complex by culture from any respiratory specimen (sputum, bronchoalveolar lavage fluid, lung tissue from biopsy or surgery), gastric aspirate, or pleural fluid [Table 1]; or (2) a positive NAAT result from a respiratory specimen together with clinical features and radiologic findings consistent with active PTB. We excluded patients who (i) had been diagnosed with PTB at other hospitals (n = 14) or (ii) had received a clinical PTB diagnosis without microbiologic confirmation using either culture or NAAT (n = 70). Patients who did not meet the PTB criteria were classified as non-PTB (n = 12,971).

Next, we applied specimen-based case-selection criteria. Initially, we excluded all patients who did not submit three consecutive sputum specimens tested by both sputum NAAT and sputum culture. After this step, 198 patients remained in the PTB group with 3,761 patients in the non-PTB group. However, of the 206 excluded PTB patients, 92 were re-included because their diagnosis had already been

microbiologically confirmed by the first or second specimen, making further sampling unnecessary. Thus, the final study population comprised 4,051 patients: 290 in the PTB group and 3,761 in the non-PTB group.”

Comment 5: *Line 94: CLSI separates pleural fluid and gastric aspirate as separate specimen types from respiratory. Change text to respiratory sources or just specimen. Also add in reference to Table 1 for specimen type breakdown.*

Response: Thank you for this valuable clarification. In accordance with the CLSI distinction between respiratory specimens and other diagnostic specimen types, we have revised the description of microbiologic confirmation in the PTB group for clarity. Also, reference has been added to Table 1 for specimen type breakdown, as recommended.

Revisions in the manuscript:

- *Methods (Page 5, Lines 95–97)*

“...any respiratory specimen (sputum, bronchoalveolar lavage fluid, lung tissue from biopsy or surgery), gastric aspirate, or pleural fluid...”

Comment 6: *Line 168-172 and 179-186: This data presentation is bulky and difficult to read through. Some suggestions are to break up into multiple sentences by test method. Also, you do not need to include the 95% CI every time. Include as "The cumulative sensitivities (95% CI) for..." which will reduce the sentence length down.*

Response: We appreciate this practical suggestion for improving readability. As suggested, we have revised both sections in the Results to reduce text density. The cumulative sensitivities are now presented in separate sentences for each test method (AFB smear, NAAT, and culture). Furthermore, we have streamlined the notation by introducing the confidence intervals once at the beginning of the sentence (e.g., "The cumulative sensitivities (95% CI) were..."), thereby avoiding repetitive phrasing and significantly reducing sentence length.

Revisions in the manuscript:

- *Results (Page 9, Line 188–Line 196)*

“Cumulative Diagnostic Yield of Consecutive Sputum Testing

In the overall PTB group (n = 290), the cumulative sensitivity of sputum AFB smear, NAAT, and culture increased with each additional sputum specimen (Table 2). For AFB smear, the cumulative sensitivities (95% CI) after one, two, and three specimens were 33.1% (27.9–38.7%), 43.1% (37.5–48.9%), and 47.9% (42.2–53.7%), respectively. For NAAT, the corresponding sensitivities were 53.1% (47.4–58.8%), 63.1% (57.4–68.5%), and 67.2% (61.6–72.4%). For culture, they were 38.3% (32.9–43.9%), 53.1% (47.4–58.8%), and 60.0% (54.3–65.5%), respectively. The specificity of both NAAT and culture remained consistently above 99% regardless of the number of specimens.

(Page 10, Lines 198–207)

Cumulative Diagnostic Yield Stratified by Sputum Smear Status

Cumulative diagnostic yield was further evaluated after stratification by patient-level sputum smear status (Table 3). In the smear-positive PTB group (n = 139), NAAT cumulative sensitivities (95% CI) for one, two, and three specimens were 84.9% (78.0–89.9%), 97.8% (94.0–99.2%), and 99.3% (95.8–99.9%), respectively. Culture sensitivities were 50.4% (42.2–58.5%), 70.5% (62.6–77.3%), and 76.3% (68.7–82.6%). In the smear-negative PTB group (n = 151), NAAT sensitivities were 23.8% (17.6–31.4%), 31.1% (24.2–38.9%), and 37.7% (30.4–45.7%). Culture sensitivities were 27.2% (20.6–34.9%), 37.1% (29.6–45.4%), and 45.0% (37.3–53.0%). Specificity for both NAAT and culture remained consistently above 99% in both subgroups, irrespective of the number of specimens.”

Comment 7: *Line 279-280: Authors could also note that liquid culture takes less time than solid media growth and may increase the time to diagnosis.*

Response: We have revised the *Limitations* paragraph to specify that all mycobacterial cultures in our study were performed using Ogawa solid medium only. The non-use of liquid culture systems, known to be more sensitive and faster, may have contributed not only to underestimation of true culture sensitivity but also to delayed microbiological confirmation. This clarification has now been incorporated into the revised text.

Revisions in the manuscript:

- *Discussion (Page 14, Lines 303–306)*

“Third, mycobacterial culture was performed exclusively on Ogawa solid medium; liquid culture systems, which are faster and more sensitive, were not employed. This may have resulted in underestimation of culture sensitivity and delayed microbiological confirmation [14].”

Comment 8: *Figure 1: please clarify either here or in text what a clinical diagnosis of PTB means for those excluded. Was this previous diagnosis, diagnosis without microbiologic testing, or something else?*

Response: For clarity, as part of the revisions addressing Comment 4, we have explicitly defined the "clinical PTB diagnosis" group in both the Study Population section and the footnote of Figure 1. In the revised text, we specify that these excluded patients were those treated for PTB based solely on clinical and radiological findings, without microbiologic confirmation by either culture or NAAT. Since our study evaluates diagnostic accuracy against a confirmed microbiologic reference standard, these unconfirmed cases were excluded from the final analysis to avoid verification bias.

Revisions in the manuscript:

- *Methods (Page 5, Lines 99–100)*

“(ii) had received a clinical PTB diagnosis without microbiologic confirmation using either culture or NAAT (n = 70)”

Comment 9: *Table 1: suggest changing "final diagnostic method" to "diagnostic specimen type". Method is generally more synonymous with testing modality than specimen type.*

Response: We agree that the term “method” is generally synonymous with testing modality rather than the specimen itself. Accordingly, we have revised the column heading in Table 1 from “Final Diagnostic Method, n (%)” to “Diagnostic Specimen Type, n (%)” as suggested.

Comment 10: *Table 2 and Table 3: suggest adding in dividing lines between the test*

methods data presentation (ie. Line across between NAAT 3 row and Culture 1 row) to allow for better visualization by readers.

Response: We agree that this update would improve readability. Tables 2 and 3 have been updated to include horizontal dividing lines between the final NAAT row (NAAT 3) and the first culture rows providing a clearer delineation between test modalities.

Re: Spectrum02442-25R1 (Incremental Diagnostic Yield of Consecutive Sputum Nucleic Acid Amplification Tests for Pulmonary Tuberculosis)

Dear Dr. Naohisa Urabe:

Your manuscript has been accepted, and I am forwarding it to the ASM production staff for publication. Your paper will first be checked to make sure all elements meet the technical requirements. ASM staff will contact you if anything needs to be revised before copyediting and production can begin. Otherwise, you will be notified when your proofs are ready to be viewed.

Sincerely,
Salika Shakir
Editor
Microbiology Spectrum